# Effects of Dietary Sodium Nitroprusside and N^G^-Nitro-L-Arginine Methyl Ester on Growth Performance, Some Organs Development Status and Immune Parameters in Broilers

**DOI:** 10.3390/ani13081361

**Published:** 2023-04-15

**Authors:** Tuba Bülbül, Şamil Sefergil, Aziz Bülbül

**Affiliations:** 1Department of Animal Nutrition and Nutritional Diseases, Milas Faculty of Veterinary Medicine, Mugla Sitki Kocman University, Mugla 48000, Türkiye; 2Department of Veterinary Anatomy, Milas Faculty of Veterinary Medicine, Mugla Sitki Kocman University, Mugla 48000, Türkiye; 3Department of Veterinary Physiology, Milas Faculty of Veterinary Medicine, Mugla Sitki Kocman University, Mugla 48000, Türkiye

**Keywords:** broiler, growth performance, nitric oxide, SNP, L-NAME

## Abstract

**Simple Summary:**

Nitric oxide (NO) plays an important role in every biological system as a gaseous hormone. NO is generated from arginine by NO synthase (NOS). NOS is inhibited by several arginine analogs, including N^G^-nitro-L-arginine methyl ester (L-NAME). NO is an important regulator of feeding behavior by suppressing feed intake (FI) as a result of modulating the appetite center through intracerebroventricular and intraperitoneal applications of NOS inhibitors. Feeding behavior can be regulated by peripheral systems. However, the effects of dietary NO donors and inhibitors on feeding behavior and performance are unknown. In this study, the aim was to evaluate the effects of dietary supplementation of sodium nitroprusside (SNP), an NO donor, and L-NAME, an NOS inhibitor, on performance and immunity. SNP suppressed FI and body weight gain in a dose-dependent manner throughout the study, especially in the initial period, and worsened the feed conversion ratio (FCR). L-NAME (100 mg/kg) increased FI and suppressed antibody titers, and L-NAME (25 mg/kg) improved the FCR in the initial period. Therefore, when formulating broiler starter diets, it is important to consider how diet composition will affect the NO metabolism, which is thought to have important effects on performance and immunity.

**Abstract:**

This study was conducted to determine the effects of dietary supplementation of sodium nitroprusside (SNP), a nitric oxide (NO) donor, and NG-nitro-L-arginine methyl ester (L-NAME), an NO synthase inhibitor, on growth performance, organ development, and immunity in broilers. A total of 560 one-day-old mixed-gender broiler chickens (ROSS 308) were divided into one control and seven experimental groups. The experimental groups were fed a basal diet supplemented with 25, 50, 100, and 200 ppm SNP, and 25, 50, and 100 ppm L-NAME in the starter and grower diets. Body weight gain increased in groups receiving 25–100 ppm L-NAME on day 21 and 100 ppm L-NAME on days 0–42. Feed intake increased in the group receiving 100 ppm L-NAME on all days. The feed conversion ratio improved in the group receiving 25 ppm L-NAME on days 0–21, whereas it worsened in groups with 100 and 200 ppm SNP on days 0–42. Serum antibody titers decreased in the 100 ppm L-NAME group on day 21. In conclusion, the supplementation of the NO synthase inhibitor L-NAME to the broilers’ diet had a positive effect on the performance parameters, whereas the NO donor SNP worsened these parameters, especially on days 0–21.

## 1. Introduction

Nitric oxide (NO) is synthesized endogenously from arginine by an enzyme with isoforms identified as neuronal, endothelial, and inducible NO synthases (nNOS/eNOS/iNOS) [1]. The NOS enzyme belongs to the cytochrome P450 protein family and is inhibited by many arginine analogs, including N^G^-nitro-L-arginine methyl ester [2]. NO is an intermediary molecule that functions like a gaseous hormone and acts on nearly every system in the body. It is believed that NO is a physiological modulator of feed intake [3]. As a signal messenger, NO plays a role in the control of feeding in many species, including mice [4], rats [5], and chickens [6,7]. It is also a central component in neuropeptide regulation of appetite [8]. Blocking NOS in the central nervous system reduces adiposity in high-fat-induced obese mice [9], and obese rats [10]. Leptin is a hormone produced by adipocytes that inhibits NO synthesis in the hypothalamus and reduces feed intake [11,12]. Intracerebroventricular (ICV) injection of L-NAME in poultry increased feed intake in laying chickens, while its intraperitoneal (IP) administration decreased feed intake in both broilers and laying chickens [6]. Moreover, it has been determined that peripherally administered L-NAME inhibits the increase in feed intake caused by a neuropeptide (Neuropeptide Y) responsible for feed intake, and NOS inhibitors may be beneficial in obesity management [13]. On the other hand, it was determined that IP administration of sodium nitroprusside (SNP), which is the exogenous source of NO, prevents feed intake in laying chickens in a dose-dependent manner [14].

The immune system, intestinal contractions, and eating habits are all regulated by NO in hens [6,7,15,16]. By increasing iNOS activity, pathogens, such as Marek’s disease virus [17], Salmonella spp. [18], and coccidial infections [19], induce NO production. Injections of lipopolysaccharide (LPS) increased the serum NO levels in chickens [20], whereas NOS inhibition decreased LPS-induced fevers [21]. Hence, NO is likely to be created in instances of chicken infection. According to reports, the fact that all tissues and cells have the enzymes necessary for NO generation explains why it has such a wide range of impacts [1,7]. Previous studies have evaluated the effects of NO donors and inhibitors with ICV [6] and with IP injections peripherally [6,14]. On the other hand, it has been determined that dietary NOS donors and inhibitors affect ovarian folliculogenesis (with the diet containing 50 and 200 mg/kg of SNP and L-NAME) in quails [22], and change the NOS expression in the jejunum (with the diet containing 25, 50, 100 and 200 mg/kg SNP, and 25, 50 or 100 mg/kg L-NAME) in chickens [15]. The gastrointestinal tract (GIT) provides the biological environment for digestion and absorption of nutrients as well as protection against pathogens and toxins. The rapid growth of broilers is due to the high absorption capacity of intestinal epithelia and the efficient conversion of nutrients to muscle. Physiologically, reactive oxygen species and reactive nitrogen species (RNS) are generated by GIT epithelial cells either from oxygen metabolism or by enteric commensal bacteria and regulated gut health. The RNS, by-products of NOS, are expressed in selected cells of the intestinal mucosa and submucosal regions. However, the overproduction of NO radicals damages the intestinal mucosa and impairs nutrient utilization [23]. In this context, it can be hypothesized that inhibition of NO by basal level L-NAME may be beneficial in terms of performance parameters. However, to the authors’ knowledge, there have been no reports concerning dietary supplementation of exogenous NO donors and inhibitors on performance and immune parameters. Therefore, the present study was designed to evaluate the effects of dietary SNP and L-NAME supplementation to broiler diets on growth performance and immunity.

## 2. Materials and Methods

### 2.1. Animals and Experimental Protocols

A total of 560 one-day-old Ross 308 hybrid mixed-gender broiler chickens were used. The chickens were randomly allocated to one control group and seven experimental groups, each containing 70 chickens. Each group was randomly divided into five replicates (pen), comprising 14 chickens in each group (7 males and 7 females). The chickens were housed in sawdust bedding, and the chicken density was 12 animals/m^2^. The house temperature was maintained at approximately 32 °C from 1 to 7 days of age, 29 °C from 8 to 14 days of age, 26 °C from 15 to 21 days of age, and 21 °C thereafter. During the experiment, the relative humidity was between 45% and 65%. For the first four days following placement, light (fluorescent, 30 lux) was provided for 23 h, and it was gradually reduced (1 h per day) to 20 h on day 7 (fluorescent, 10 lux). Feed and water were provided ad libitum. A vaccination program for broilers was designed as the following: day 0 with inactive Infectious Bursal Disease (IBD) + Newcastle Disease (ND) vaccine (Gumbopest, Merial RTA, subcutaneously), day 7 with live ND vaccine (Nobilis ND Lasota, Intervet, in drinking water) and infectious bronchitis vaccine (Nobilis, Intervet, in drinking water), day 14 with live IBD vaccine (Bursine Plus, Ford Dodge-Refarm, in drinking water), and day 21 with live ND vaccine (Nobilis ND Lasota, Intervet, in drinking water).

All broilers were fed corn, wheat, corn gluten, soybean meal, sunflower meal, full-fat soybean and a blood meal-basal diet that contained the critical nutrients recommended by the NRC (1994) without added antibiotics, coccidiostats or growth promoters for up to 42 days. From 0 to 21 days of age, they received a starter diet (22.07% crude protein; 3200 kcal/kg metabolizable energy), and from 22 to 42 days of age, they received a grower diet (19.86% crude protein; 3200 kcal/kg metabolizable energy), as shown in Table 1. The nutrient composition of the basal diets, including dry matter, crude protein, crude fat, crude fiber, and crude ash contents, was determined according to the AOAC (2000). Metabolizable energy, including calcium, phosphorus, arginine, lysine, methionine, cysteine and methionine contents, were determined as described by Jurgens [24].

The control group was fed the basal diet throughout the experiment, whereas the experimental groups were fed the basal diet supplemented with 25, 50, 100, and 200 mg/kg SNP (S0501; Sigma-Aldrich Chemical Co., St. Louis, MO, USA), and 25, 50, and 100 mg/kg of L-NAME (S5501; Sigma).

### 2.2. Growth Performance

The body weight (BW) of each animal was recorded per floor pen on days 1, 21, and 42. Feed intake and BW gain of birds in each pen were recorded for days 0–21 and 22–42. The feed conversion ratio (FCR) was calculated by dividing the cumulative feed intake per floor pen by the body mass per pen at the end of the measurements after 21 and 42 days. Mortality was recorded daily, and feed intake was corrected afterwards.

### 2.3. Organ Development

At the end of days 21 and 42, 10 birds (different genders, two birds per replicate) from each group that were closest to the mean body weight of the group average were selected, and these 160 birds were sacrificed. The heart, liver, bursa of Fabricius, spleen, proventriculus, and gizzard were removed. The ratio 100 × (organ weights (g)/BW (g)) was used to calculate the relative organ weights.

### 2.4. Immune Response Parameters

At the end of the starter and grower periods, blood samples were taken from wing veins into sterile tubes with or without anticoagulant (heparin) from 10 birds randomly selected from the groups (two birds per replicate). After clotting at room temperature for 1 h and centrifugation (3000 rpm, 15 min), the serum was carefully harvested. The infectious bursal disease (IBD) antibody titer was determined by ELISA with a commercial test kit (IBD ELISA kit, Bio-check Company, Reeuwijk, The Netherlands) according to the manufacturer’s instructions in an ELISA reader. The heterophil/lymphocyte (H/L) ratio was calculated from 100 cells per slide and classified using oil immersion microscopy at 100×objective. WBC counts were done by hemocytometer, using a quantity of blood samples mixed with the diluent (Natt-Herricks Solution) [24].

### 2.5. Serum Biochemistry Parameters

Serum NO concentrations were determined according to the procedure of Miranda [25]. Nitrate was reduced to nitrite with vanadium (III) and the nitrite level was measured by using Griess reagents. The serial dilutions 0.5–200 μM of sodium nitrate (Merck, Darmstadt, Germany) were used as standards. The results were expressed as μM. Serum total protein, alkaline phosphatase (ALP), alanine aminotransferase (ALT), aspartate aminotransferase (AST), creatinine, and urea were measured with an auto-analyzer (Tokyo Boeki Prestige 24i, Kyobashi, Japan).

### 2.6. Statistical Analyses

The SPSS for Windows General Linear Models procedure was used to analyze the data from treatment means in a completely randomized design. For performance data, pen means served as the experimental unit for statistical analysis. All data were tested for normality using the Shapiro–Wilk test and Levene’s test to examine the homogeneity of variance. When differences (*p* < 0.05) among means were found, means were separated using Tukey’s studentized range test. Linear and nonlinear SNP and L-NAME dose-response curves were plotted using the GLM procedure of SPSS. The differences were considered significant at *p* < 0.05.

## 3. Results

### 3.1. Growth Performance

The effects of SNP and L-NAME on growth performance in broilers are presented in Table 2. SNP supplementation had a linear effect on BW gain, FCR (on days 0–21, 22–42, and 0–42), and feed intake (on days 0–21 and 0–42). However, L-NAME in the diet created a nonlinear effect on BW gain (on days 0–21) and feed intake (on days 0–21, 22–42, and 0–42), as well as a nonlinear effect on feed intake and the FCR (on day 0–21). BW gain decreased in groups that were fed the diet containing 50, 100, and 200 ppm SNP on days 0–21 (*p* < 0.001) and 22–42 (*p* < 0.01), and in the group receiving 100 and 200 ppm SNP on day 0–42 (*p* < 0.01). BW gain increased in groups that were fed the diet containing 25, 50, and 100 ppm L-NAME on day 21 (*p* < 0.001) and 100 ppm L-NAME on days 0–42 (*p* < 0.01). Feed intake decreased in the group that was fed the diet containing 100 and 200 ppm SNP on days 0–21 (*p* < 0.001), and 200 ppm SNP on days 0–42 (*p* < 0.01). However, it increased in dietary L-NAME by 100 ppm on days 0–21 (*p* < 0.001), 50 ppm on days 22–42 (*p* < 0.01), and 100 ppm on days 0–42 (*p* < 0.01). FCR improved in the group that was fed the diet containing 25 ppm L-NAME (*p* < 0.001) on days 0–21, whereas it worsened in the diet supplemented with 100 and 200 ppm SNP on days 0–21 (*p* < 0.001), 22–42 (*p* < 0.01) and 0–42 (*p* < 0.001).

### 3.2. Organ Traits

It was detected that the relative weight of bursa of Fabricius showed a linear effect in the SNP groups and a nonlinear effect in the L-NAME groups. The relative weight of bursa of Fabricius increased in groups that were fed the diet containing 25, 100, and 200 ppm SNP on day 21, whereas it decreased in the 100 ppm L-NAME group (*p* < 0.05; Table 3). The relative weights of the heart, liver, spleen, proventriculus, and gizzard did not show any significant difference between groups (Table 3).

### 3.3. Immune Response Parameters

It was observed that the IBD antibody titer decreased in the group that was fed the diet containing 100 ppm L-NAME on day 21 (*p* < 0.01). On day 21, the H/L ratio increased in the group that was fed the diet containing 200 ppm SNP (*p* < 0.05). The WBC ratio did not differ significantly between periods (Table 4).

### 3.4. Serum Biochemistry Parameters

In the current study, serum NOx level increased with dietary 50, 100, and 200 ppm SNP on day 21 and 200 ppm SNP on day 42, whereas 50 and 100 ppm L-NAME supplementation caused a decrease in serum NOx level (*p* < 0.01). Serum total protein, ALP, ALT, AST, creatinine, and urea were not affected by supplementation with SNP or L-NAME (*p* > 0.05; Table 5).

## 4. Discussion

Sodium nitroprusside is widely used as an exogenous NO donor, especially to investigate the efficacy of NO in in vitro and in vivo studies [14,22]. Although NO can be measured in many direct and indirect ways, the short half-life of NO reduces the practicality of these methods for the evaluation of in vivo biological samples. It is also stated that these procedures are generally not suitable for clinical laboratories due to instrumentation requirements and inconvenience in handling large numbers of samples. Therefore, the difficulties in determining the amount of NO are eliminated by measuring the stable end products, especially nitrite (NO_2_) and nitrate (NO_3_) [25]. In this study, NO metabolism was determined by the serum NOx levels. It was observed that serum NOx levels increased (*p* < 0.01) when SNP was added to the diet at levels of 50 mg/kg and above on days 0–21 and 200 mg/kg on days 22–42. L-arginine analogs such as L-NAME act as NOS inhibitors because of their displacement in one or both of the terminal guanidino (G or w) nitrogen atoms [25]. In contrast to SNP, the supplementation of L-NAME to the diets at levels of 50 and 100 mg/kg decreased (*p* < 0.01) the serum NOx level numerically at the beginning of the study (days 0–21) and statistically throughout the study (Table 5).

In this study, it was determined that SNP added to the diet suppressed BW gain and feed intake in all periods, especially on days 0–21 (*p* < 0.001; Table 2). It is known that the ICV injection of NO donors suppresses feed intake and shows this effect by mediating the release of neuropeptides, which are involved in the regulation of the appetite and satiety centers in the hypothalamus [6]. On the other hand, the efficiency of the appetite and satiety centers can also be regulated by peripheral signals. Indeed, peripheral signals perform actions through the afferent neuron and brainstem, which will indirectly affect the hypothalamus. Mechanoreceptors and/or chemoreceptors also contribute to the control of the appetite [26,27]. In fact, the suppression of intestinal contractions reduces feed intake by suppressing the stimulation of the appetite center [28]. In our previous research [15], we showed that the expression of nNOS, which is responsible for the release of NO and inhibits contractions in the intestinal tissue, increased with the supplementation of 50–200 ppm SNP to the diet, whereas it was suppressed by 100 ppm L-NAME. Again, we found that L-arginine, an endogenous donor, and SNP, an exogenous donor of NO, inhibit the contractions of the small and large intestines in vivo and in vitro [15,16]. In the current study, unlike SNP, we demonstrated that L-NAME, an NOS inhibitor, increased feed intake in all periods, especially on days 0–21, and it also increased BW gain in the initial period and throughout the study. It can be assumed that the decreased intestinal contraction in the SNP groups suppressed the appetite center peripherally and then decreased the feed intake. This finding supports the previous observations that IP administration of SNP to chickens [14] and quails [29] decreases feed intake in a dose-dependent manner.

The results clearly showed that the supplementation of 100 and 200 mg/kg of SNP to the diet worsened the FCR on days 0–21 (*p* < 0.001), 22–42 (*p* < 0.01), and 0–42 (*p* < 0.001), whereas the FCR improved in the group in which the diet contained 25 mg/kg L-NAME on days 0–21 (*p* < 0.001; Table 2). NO is responsible for the regulation of absorption, secretion, and motility in the gastrointestinal tract [15,30]. Absorption and smooth muscle activity of the small intestine are the main factors that promote and regulate the transport and absorption of nutrients [31]. In this context, the worsening of the FCR by SNP may be due to impaired intestinal contractions, secretion, and/or absorption after increased NOS enzyme activity. On the other hand, the supplementation of 25 mg/kg L-NAME to the diet, which causes inhibition of the NOS enzyme responsible for the release of a molecule that is functional in the digestive tract such as NO, the whole body had a positive effect on the FCR (*p* < 0.001; Table 2). It is quite difficult to explain this situation with the parameters examined in the research. However, NO is also a free radical and increases lipid peroxidation because it carries an unpaired electron in its free orbit. Therefore, it is considered that a partial decrease in the amount of NO may lead to an improvement in the FCR (Table 5).

While performance parameters worsened in the SNP groups, there was no difference in mortality rates between the groups, hence the need to evaluate liver and kidney enzyme levels (Table 5). In mammals and poultry, the enzymes ALP, ALT and AST change during liver damage, while kidney damage leads to changes in creatine. However, there was no difference in the liver and kidney enzyme levels in the control and experimental groups of this study. Therefore, it is determined that the supplementation of SNP and L-NAME does not cause liver or kidney damage. There was also no change in serum protein levels in this study. The reason for the increase in the protein concentration in the serum is related to protein degeneration, dietary protein intake, or the metabolism of orally ingested protein in the liver, while its decrease is associated with the diet as well as liver and kidney damage [32]. This showed that the decrease in performance was not due to organ and/or tissue damage.

Nitric oxide donors are involved in the regulation of immunity as well as performance due to their capacity to act as a substrate for NO [20]. Studies have shown that blood NO level and NOS expiration change as a result of various viral, bacterial, and protozoan diseases in poultry [18,20]. In this study, the effects of the dietary supplementation of donors and inhibitors on NO metabolism were evaluated by measuring the lymphoid tissue weight, WBC count, and IBD titers after vaccination. It is known that the shape and size of the lymphoid organs are associated with animal’s health status [33]. The bursa of Fabricius is a primary lymphoid organ in birds that plays an important role for the maturation of B cells. It has a distinctive anatomical structure and regulates the total number of leukocytes and lymphocytes through differentiation and proliferation of B cells [33,34]. In studies conducted in broilers [35,36,37], it has been reported that insufficiency of L-arginine, an NO donor, decreases the relative weight of the bursa of Fabricius. However, the diet supplemented with arginine above the level described in the NRC did not affect the relative weight of bursa of Fabricius [38,39]. In the present study, the relative weight of bursa of Fabricius increased in SNP groups on days 0–21, whereas it decreased in the 100 ppm L-NAME group (*p* < 0.05; Table 3). The relative weights of the heart, liver, spleen, proventriculus, and gizzard in broilers were not affected by the levels of SNP or L-NAME in the diets (Table 3).

Stress can be defined as an adaptive response to threats that threaten a bird’s homeostasis [40]. Stress factors include light, temperature, air quality, environmental pollutants, feed composition, and physio-pathological changes [41]. The increase in the percentage of heterophiles with unchanged WBC in the blood represents stress in chickens and quails [42,43]. In this study, the decrease in growth performance, especially with the supplementation of 200 mg/kg SNP to the diet, suggests that stress may have an additional effect as well on feed intake.

In this study, it was observed that the serum IBD antibody titer at day 21 was lower in the group supplemented with 100 mg/kg L-NAME (*p* < 0.01; Table 4). In addition, it was observed that the H/L ratio decreased in this group and other L-NAME groups, although it was not statistically significant (*p* > 0.05). Considering that nitric oxide modulates the circulating lymphocyte subpopulation and inflammatory cytokine expression in chickens [44,45], the reduction in IBD titer suggests that L-NAME may suppress IBD titer by affecting lymphoid tissues or inhibiting lymphocytes.

## 5. Conclusions

As a result, it was observed that the increase in NO metabolism, especially on days 0–21, had a negative effect on growth performance by decreasing feed intake and suppressing BW gain. On the other hand, although the inhibition of NOS enzyme seems to improve the FCR in a dose-dependent manner, it is considered that there is a need for detailed studies on the immune system. Therefore, the effects of the composition of the diets on NO metabolism, which has important effects on performance and immunity, should be considered, especially when preparing broiler starter diets.

## Figures and Tables

**Table 1 animals-13-01361-t001:** Ingredients and chemical composition of the basal diets (g/kg).

	Starter (0–21 Days)	Grower (22–42 Days)
Ingredients		
Corn	497.54	493.59
Wheat	100.00	150.00
Corn gluten	120.00	107.33
Soybean meal	147.13	95.06
Sunflower meal	26.83	53.64
Full-fat soybean	21.02	22.30
Blood meal	30.00	20.00
Vegetable oil	30.00	30.00
Limestone	6.15	9.58
Dicalcium phosphate	9.97	6.92
Salt	2.70	2.82
DL-Methionine	0.20	-
L-Lysine hydrochloride	2.71	3.01
Sodium bicarbonate	2.25	2.25
Vitamin premix ^a^	2.50	2.50
Mineral premix ^b^	1.00	1.00
Analysis results		
Dry matter	918.60	917.50
Crude protein	220.70	198.60
Crude fat	67.70	66.40
Crude fibre	28.30	31.30
Crude ash	54.00	50.20
Calculation results ^c^		
Calcium	10.00	9.00
Available phosphorus	4.50	3.50
Arginine	12.50	11.00
Lysine	11.00	10.00
Methionine	5.00	4.50
Methionine+cysteine	9.15	8.34
Metabolizable energy, kcal × kg^−1^	3200	3200

^a^ Provides per kg diet: Trans-retinol 12,000 IU, cholecalciferol 1500 IU, α-tocopherol acetate 75 mg, thiamin 3 mg, riboflavin 6 mg, pyridoxine 5 mg, cobalamin 0.03 mg, nicotineamide 40 mg, panthotenic acid 10 mg, folic acid 0.75 mg, choline 375 mg, and biotin 0.075 mg; ^b^ Provides per kg diet: Mn 80 mg, Fe 40 mg, Zn 60 mg, Cu 5 mg, I 0.5 mg, Co 0.2 mg, and Se 0.15 mg; ^c^ Calculated by using values in the table [24].

**Table 2 animals-13-01361-t002:** Effect of dietary sodium nitroprusside (SNP) and N^G^-nitro-L-arginine methyl ester (L-NAME) supplementation on body weight gain, feed intake, and feed conversion ratio on days 0–21, 22–42, and 0–42 days.

	Body Weight Gain (g)			Feed Intake (g)			Feed Conversion Ratio		
	0–21 Days	22–42 Days	0–42 Days	0–21 Days	22–42 Days	0–42 Days	0–21 Days	22–42 Days	0–42 Days
Control	889.1 ^b^	1746 ^ab^	2635 ^bc^	1372 ^b^	3463 ^bc^	4836 ^bc^	1.544 ^cd^	1.984 ^c^	1.835 ^cd^
SNP25	862.3 ^bc^	1691 ^bc^	2554 ^bc^	1323 ^bc^	3502 ^abc^	4826 ^bc^	1.535 ^cd^	2.074 ^bc^	1.891 ^cd^
SNP50	832.9 ^c^	1628 ^cd^	2461 ^cd^	1333 ^b^	3387 ^c^	4720 ^cd^	1.601 ^bc^	2.082 ^bc^	1.918 ^bc^
SNP100	779.3 ^d^	1579 ^d^	2359 ^d^	1273 ^cd^	3432 ^c^	4706 ^cd^	1.637 ^b^	2.187 ^ab^	2.003 ^b^
SNP200	717.0 ^e^	1476 ^e^	2193 ^e^	1236 ^d^	3409 ^c^	4646 ^d^	1.725 ^a^	2.309 ^a^	2.118 ^a^
LN25	924.5 ^a^	1802 ^a^	2727 ^a^	1338 ^b^	3562 ^ab^	4901 ^ab^	1.448 ^e^	1.976 ^c^	1.797 ^d^
L-N50	926.8 ^a^	1720 ^abc^	2647 ^ab^	1357 ^b^	3587 ^a^	4945 ^ab^	1.465 ^de^	2.08 ^bc^	1.869 ^cd^
L-N100	928.8 ^a^	1756 ^ab^	2685 ^ab^	1449 ^a^	3584 ^a^	5033 ^a^	1.561 ^bc^	2.043 ^bc^	1.876 ^cd^
Pooled SEM	12.17	19.32	27.30	10.30	17.86	24.69	0.01	0.02	0.02
*p*<	0.001	0.001	0.001	0.001	0.001	0.001	0.001	0.01	0.001
*	0.001	0.001	0.001	0.001	-	0.01	0.001	0.001	0.001
**	0.05	-	-	0.01	0.05	0.01	-	-	-
***	-	-	-	-	-	-	-	-	-
****	-	-	-	0.01	-	-	0.01	-	-

^a,b,c,d,e^ Mean values within the same row sharing a common superscript letter are not statistically different at *p* < 0.05; *: Linear effect in SNP groups; **: Linear effect in L-NAME groups; ***: Nonlinear effect in SNP groups; ****: Nonlinear effect in L-NAME groups.

**Table 3 animals-13-01361-t003:** Effect of dietary sodium nitroprusside (SNP) and N^G^-nitro-L-arginine methyl ester (L-NAME) supplementation on relative organ weights (%) on days 0–21 and 22–42.

			SNP, mg/kg				L-NAME, mg/kg			Pooled SEM	*p*<	Linear		Nonlinear	
Item	Days	Control	25	50	100	200	25	50	100			*	**	***	****
Heart	21	0.98	1.09	0.93	0.94	1.01	0.94	0.95	0.94	0.019	-	-	-	-	-
	42	0.81	0.86	0.74	0.88	0.75	0.91	0.87	0.81	0.016	-	-	-	-	-
Liver	21	3.09	3.18	3.12	3.18	3.28	3.24	3.23	3.33	0.059	-	-	-	-	-
	42	2.76	2.60	2.84	2.45	2.65	2.56	2.95	2.47	0.076	-	-	-	-	-
Bursa of Fabricius	21	0.20 ^b^	0.25 ^a^	0.22 ^ab^	0.22 ^ab^	0.22 ^ab^	0.23 ^ab^	0.19 ^ab^	0.15 ^c^	0.007	0.004	0.042	-	-	0.038
	42	0.15	0.20	0.20	0.17	0.17	0.19	0.18	0.13	0.028	-	-	-	-	-
Spleen	21	0.10	0.099	0.09	0.09	0.097	0.096	0.10	0.11	0.003	-	-	-	-	-
	42	0.13	0.15	0.12	0.14	0.17	0.14	0.13	0.15	0.006	-	-	-	-	-
Proventriculus	21	0.80	0.81	0.83	0.79	0.81	0.78	0.75	0.76	0.01	-	-	-	-	-
	42	0.40	0.48	0.527	0.50	0.47	0.43	0.43	0.48	0.001	-	-	-	-	-
Gizzard	21	3.17	2.94	3.29	3.65	3.56	2.71	3.21	3.18	0.08	-	-	-	-	-
	42	2.04	2.03	2.21	2.20	2.15	1.71	1.92	2.09	0.05	-	-	-	-	-

^a,b,c^ Mean values within the same row sharing a common superscript letter are not statistically different at *p* < 0.05; *: Linear effect in SNP groups; **: Linear effect in L-NAME groups; ***: Nonlinear effect in SNP groups; ****: Nonlinear effect in L-NAME groups.

**Table 4 animals-13-01361-t004:** Effect of dietary sodium nitroprusside (SNP) and N^G^-nitro-L-arginine methyl ester (L-NAME) supplementation on some immune response parameters.

			SNP, mg/kg				L-NAME, mg/kg			Pooled SEM	*p*<	Linear		Nonlinear	
Item	Days	Control	25	50	100	200	25	50	100			*	**	***	****
IBD mean titer	d 21	624.12 ^a^	526.00 ^a^	644.87 ^a^	542.50 ^a^	588.00 ^a^	596.62 ^a^	583.37 ^a^	350.12 ^b^	19.77	0.01	-	-	0.01	0.08
	d 42	8500	9567	9179	9565	8735	9461	9796	9054	156.09	-	-	-	-	-
WBC 10^3^/mL	d 21	1806	1766	1785	1776	1685	1730	1782	1715	17.16	-	-	-	-	-
	d 42	1651	1636	1631	1677	1612	1686	1601	1642	15.43	-	-	-	-	-
H/L	d 21	0.606 ^b^	0.553 ^b^	0.595 ^b^	0.582 ^b^	0.723 ^a^	0.603 ^b^	0.561 ^b^	0.544 ^b^	0.01	0.05	0.05	0.01	-	-
	d 42	0.613	0.639	0.654	0.650	0.683	0.613	0.631	0.581	0.01	-	-	-	-	-

IBD: Infectious bursal disease, WBC: White blood cell, H/L: Ratio of heterophils and lymphocytes; ^a,b^ Mean values within the same row sharing a common superscript letter are not statistically different at *p* < 0.05; *: Linear effect in SNP groups; **: Linear effect in L-NAME groups; ***: Nonlinear effect in SNP groups; ****: Nonlinear effect in L-NAME groups.

**Table 5 animals-13-01361-t005:** Effect of dietary sodium nitroprusside (SNP) and N^G^-nitro-L-arginine methyl ester (L-NAME) supplementation on some serum biochemistry parameters.

			SNP, mg/kg				L-NAME, mg/kg			Pooled SEM	*p*<	Linear		Nonlinear	
Item	Days	Control	25	50	100	200	25	50	100			*	**	***	****
NOx, µM	21	14.12 ^c^	16.73 ^bc^	20.61 ^ab^	22.84 ^a^	25.68 ^a^	11.03 ^c^	13.51 ^c^	12.80 ^c^	0.98	0.01	-	-	-	-
	42	12.91 ^b^	13.71 ^b^	13.55 ^b^	13.80 ^b^	17.05 ^a^	12.62 ^bc^	8.77 ^c^	10.40 ^c^	0.495	0.01	-	-	-	-
T. Prot, g/dL	21	3.493	3.306	3.518	3.548	3.225	3.666	3.363	3.252	0.053	-	-	-	-	-
	42	3.448	3.705	3.513	3.403	3.402	3.575	3.391	3.421	0.048	-	-	-	-	-
ALP, U/L	21	1291	1288	1434	1428	1369	1427	1385	1434	20.95	-	-	-	-	-
	42	1386	1469	1503	1502	1446	1401	1435	1437	22.32	-	-	-	-	-
ALT, U/L	21	4.250	4.50	4.250	5.125	4.125	4.750	4.750	4.500	0.188	-	-	-	-	-
	42	4.125	4.25	4.125	3.625	4.875	4.250	4.000	4.500	0.186	-	-	-	-	-
AST, U/L	21	210.42	215.00	237.75	239.37	266.00	213.75	268.25	255.75	7.47	-	-	-	-	-
	42	181.50	197.00	230.75	188.37	185.85	186.25	172.75	172.25	5.40	-	-	-	-	-
Cre, mg/dL	21	0.052	0.061	0.057	0.080	0.072	0.072	0.056	0.067	0.002	-	-	-	-	-
	42	0.054	0.040	0.055	0.045	0.052	0.042	0.055	0.053	0.001	-	-	-	-	-
Urea, mg/dL	21	3.413	3.707	3.267	3.505	3.217	3.300	3.271	3.145	0.08	-	-	-	-	-
	42	2.447	2.888	3.050	2.871	2.828	2.370	2.128	2.265	0.09	-	-	-	-	-

NOx—Nitrite plus nitrate, T. Prot—Total protein, ALP—Alkaline phosphatase, ALT—Alanine transaminase, AST—Aspartate transaminase, Cre—creatinine; ^a,b,c^ Mean values within the same row sharing a common superscript letter are not statistically different at *p* < 0.05; *: Linear effect in SNP groups; **: Linear effect in L-NAME groups; ***: Nonlinear effect in SNP groups; ****: Nonlinear effect in L-NAME groups.

## Data Availability

The raw data supporting the results of this paper will be made accessible without restriction by the authors.

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
