# Peer review of "Effects of Dietary Sodium Nitroprusside and N^G^-Nitro-L-Arginine Methyl Ester on Growth Performance, Some Organs Development Status and Immune Parameters in Broilers"

_animals, 2023, doi:10.3390/ani13081361_

Round 1

Reviewer 1 Report

The authors demonstrated that in broilers, dietary supplementation of nitric oxide (NO) donor sodium nitroprousside (SNP) had negative impact on body weight gain, feed intake and feed conversion ratio, but improved lymphoid organ development at high dose, while supplementation of NO synthase inhibitor NG-nitro-L-arginine methyl ester (L-NAME) improved body weight gain, food intake and feed conversion ratio, as well as some immune response parameters at high dose during the first 21 days of life. This study provided insights in using dietary NO supplements for manipulating growth performance and immunity, and illustrated the importance of NO balance for broiler health. The study was well designed and the manuscript was well organized, presented and written. It would be better if the authors could further analyze the sex differences and provide more time points for dose-dependent analyses.

Author Response

Reviewer I

The authors demonstrated that in broilers, dietary supplementation of nitric oxide (NO) donor sodium nitroprousside (SNP) had negative impact on body weight gain, feed intake and feed conversion ratio, but improved lymphoid organ development at high dose, while supplementation of NO synthase inhibitor NG-nitro-L-arginine methyl ester (L-NAME) improved body weight gain, food intake and feed conversion ratio, as well as some immune response parameters at high dose during the first 21 days of life. This study provided insights in using dietary NO supplements for manipulating growth performance and immunity, and illustrated the importance of NO balance for broiler health. The study was well designed and the manuscript was well organized, presented and written. It would be better if the authors could further analyze the sex differences and provide more time points for dose-dependent analyses.

We would like to thank the referee for these important observations and evaluations.

If the current research could be designed to reveal the effect of gender on the parameters studied (parameter tracking by different sex), the data to be obtained would directly reveal gender-based details in terms of performance and immune system. This is a very valuable criticism. However, the reason why the current research was carried out on mixed sex is because of the advantages of making a planning for the production practice, that is, for the existing commercial broiler production models (mixed sex), or to obtain data for the field. For this reason, it was prioritized to optimize the effect of gender by keeping equal numbers of males and females in the groups.We believe that it is absolutely necessary to carry out this research as future studies with plans in which it is evaluated in terms of gender difference.

Similarly, follow-up of more doses is a suggestion that will allow the results to be evaluated more effectively. However, the fact that the ethical committees will not allow the number of animals to increase has led us to this limited design.

Reviewer 2 Report

It is a new topic in poultry nutrition and metabolic disorders.

Main questions:

Is there difference between male and female chicks in response to NO donor and inhibitors?

The authors did not separate the sexes.

Using Feed instead of food is preferred in the animal science manuscript from terminology view point.

The authors measured the NO in the blood, but we know that NO immediately reduced to nitrite. Therefore, it should be clarified, how they measured the NO.

What is the basis for choosing NO and NO inhibitor dosage? Need more explanation.

Possibly, in the future, this type of compounds uses as a therapeutic. So more information and many researches are needed.

The specific improvements should the authors consider regarding the methodology:

Separate analysis of data for male and female, or using a factorial arrangement.

Review and amendment of the measurement of NO in blood. I believe that they should report the data about nitrite instead of NO. Again, because NO is unstable in the blood and its detection is difficult and infeasible per se.

The main question posed will be addressed after correction in the methods used and results for NO or at least explain about method of measurement for increasing the confidence of the reader to the data.

The references are appropriate.

Please see more comments on the text.

Author Response

Reviewer II

It is a new topic in poultry nutrition and metabolic disorders.

Main questions:

Is there difference between male and female chicks in response to NO donor and inhibitors?

The authors did not separate the sexes.

Thanks for the valuable insights and comments.

If the current research could be designed to reveal the effect of gender on the parameters studied (parameter tracking by different sex), the data to be obtained would directly reveal gender-based details in terms of performance and immune system. This is a very valuable criticism. However, the reason why the current research was carried out on mixed sex is because of the advantages of making a planning for the production practice, that is, for the existing commercial broiler production models (mixed sex), or to obtain data for the field. For this reason, it was prioritized to optimize the effect of gender by keeping equal numbers of males and females in the groups. We believe that it is absolutely necessary to carry out this research as future studies with plans in which it is evaluated in terms of gender difference.

Using Feed instead of food is preferred in the animal science manuscript from terminology view point:  corrected as suggested. 

The authors measured the NO in the blood, but we know that NO immediately reduced to nitrite. Therefore, it should be clarified, how they measured the NO. Please see other explanations about nitric oxide below*.

 What is the basis for choosing NO and NO inhibitor dosage? Need more explanation. corrected as suggested. (L 73-75).

Possibly, in the future, this type of compounds uses as a therapeutic. So more information and many researches are needed.

The following paragraph was added to the introduction in line with the reviewer's suggestion. (L 75-89).

The gastrointestinal tract (GIT) provides the biological environment for digestion and absorption of nutrients as well as protection against pathogens and toxins. The rapid growth of broilers is due to the high absorption capacity of intestinal epithelia and the efficient conversion of nutrients to the muscle. Physiologically, reactive oxygen species (ROS) and reactive nitrogen species (RNS) are generated by GIT epithelial cells either from oxygen metabolism or by enteric commensal bacteria and regulate gut health. Reactive nitrogen species (RNS), by-products of NOS, are expressed in selected cells of the intestinal mucosa and submucosal regions. However, the overproduction of nitric oxide radicals damages the intestinal mucosa and impairs nutrient utilization. In this context, it can be hypothesized that inhibition of NO by basal level L-NAME may be beneficial in terms of performance parameters. However, to the authors’ knowledge, there have been no reports concerning dietary supplementation of exogenous NO donors and inhibitors on performance and immune parameters. Therefore, the present study was designed to evaluate the effects of dietary SNP and L-NAME supplementation to broiler diets on growth performance and immunity.

The specific improvements should the authors consider regarding the methodology:

Separate analysis of data for male and female, or using a factorial arrangement.

In terms of performance parameters, it is not possible to separate and reanalyze the data on the basis of gender. Because the study was not designed in this plan. However, organ weights, immune response parameters, and serum biochemical parameters can be analyzed and included in the main text, taking into account the gender factorial arrangement.

Review and amendment of the measurement of NO in blood. I believe that they should report the data about nitrite instead of NO. Again, because NO is unstable in the blood and its detection is difficult and infeasible per se.

The main question posed will be addressed after correction in the methods used and results for NO or at least explain about method of measurement for increasing the confidence of the reader to the data.

*As the regarded reviewer suggested, although nitric oxide can be measured in many direct and indirect ways (for example, gas and liquid chromatography, electron paramagnetic resonance, mass spectrometry, spectrophotometry, and electrochemistry), its short half-life and low NO concentrations complicate the evaluation of biological samples in vivo. reduces the applicability of the methods. In addition, these procedures are often not suitable for the clinical laboratory due to instrumentation requirements and the difficulty of handling large numbers of samples. Difficulties in quantifying NO can be eliminated by measuring its stable metabolites, particularly nitrite (NO2) and nitrate (NO3). In this study, this was preferred, and the material was also expressed in the method.  (L157-160)

Therefore, the first paragraph of the discussion was arranged as follows. The results of NO in the text are expressed as NOx

*“Sodium nitroprusside is widely used as an exogenous NO donor, especially to investigate the efficacy of NO in in vitro and in vivo studies [14,22]. Although NO can be measured in many direct and indirect ways, the short half-life of NO reduces the practicality of these methods for the evaluation of in vivo biological samples. It is also stated that these procedures are generally not suitable for the clinical laboratory due to instrumentation requirements and inconvenience in handling large numbers of samples. Therefore, the difficulties in determining the amount of NO are eliminated by measuring the stable end products, especially nitrite (NO2) and nitrate (NO3) (24) In this study, NO metabolism was determined by serum NOx level. It was observed that serum NOx levels increased (p < 0.01) when SNP was added to the diet at levels of 50 mg/kg and above on days 0–21 and 200 mg/kg on days 22–42. L-arginine analogues such as L-NAME act as NOS inhibitors, because of their displacement in one or both of the terminal guanidino (G or w) nitrogen atoms [25]. In contrast to SNP, the supplementation of L-NAME to the diets at levels of 50 and 100 mg/kg decreased (p < 0.01) serum NO level numerically at the beginning of the study (days 0–21) and statistically throughout the study (Table 5).”(L 235-241)

The references are appropriate.

Please see more comments on the text.

Reviewer 3 Report

Overall, this manuscript is well written and executed. The results are interesting and may be relevant to the industry. However, it has some limitations especially with the experimental design. The use of 5 replicates per treatment is low. 

Abstract:

Please include the design and approach you used for vaccination. The immunity part is not clear. 

Introduction:

Please add a feasibility statement about the use of L-NAME in poultry production. Can it be used regularly? 

L16: suppressing 

Materials and Methods:

L82-83: Please specify if the 14 chickens were housed in a single pen. Also, specify if you had applied any type of randomization of the pens. 

L82-83: 5 replicates per treatment seems too low. What was the rationale for using this low replication? 

L96: remove "including"

L116-119: It's not clear if 10 chickens were sacrificed for each day 21 and 28. 

L120-130: It's not clear if ad when you vaccinated them with IBD. Please specify

Author Response

Reviewer III

We would like to thank the referee for these important observations and evaluations.

Overall, this manuscript is well written and executed. The results are interesting and may be relevant to the industry. However, it has some limitations especially with the experimental design. The use of 5 replicates per treatment is low. 

Abstract:

Please include the design and approach you used for vaccination. The immunity part is not clear.  The following text has been added to the material and method.

“A vaccination program for broilers was designed as day 0 with inactive Infectious Bursal Disease (IBD) + Newcastle Disease (ND) vaccine (Gumbopest, Merial RTA, subcutaneously), day 7 with live ND vaccine (Nobilis ND Lasota, Intervet, in drinking water) and infectious bronchitis vaccine (Nobilis, Intervet, in drinking water), day 14 with live IBD vaccine (Bursine Plus, Ford Dodge-Refarm, in drinking water), and day 21 with live ND vaccine (Nobilis ND Lasota, Intervet, in drinking water) vaccines.” (L108-113)

Introduction:

Please add a feasibility statement about the use of L-NAME in poultry production. Can it be used regularly? 

The following text has been added to the introduction.

“The gastrointestinal tract (GIT) provides the biological environment for digestion and absorption of nutrients as well as protection against pathogens and toxins. The rapid growth of broilers is due to the high absorption capacity of intestinal epithelia and the efficient conversion of nutrients to muscle. Physiologically, reactive oxygen species (ROS) and reactive nitrogen species (RNS) are generated by GIT epithelial cells either from oxygen metabolism or by enteric commensal bacteria and regulate gut health. Reactive nitrogen species (RNS), by-products of nitric oxide synthases (NOS), are expressed in selected cells of the intestinal mucosa and submucosal regions. However, the overproduction of nitric oxide radicals damages the intestinal mucosa and impairs nutrient utilization. In this context, it can be hypothesized that inhibition of NO by basal level L-NAME may be beneficial in terms of performance parameters. However, to the authors’ knowledge, there have been no reports concerning dietary sup-plementation of exogenous NO donors and inhibitors on performance and immune pa-rameters. Therefore, the present study was designed to evaluate the effects of dietary SNP and L-NAME supplementation to broiler diets on growth performance and immunity.” (L75-89)

L16: suppressing: corrected as suggested. 

Materials and Methods:

L82-83: Please specify if the 14 chickens were housed in a single pen. Also, specify if you had applied any type of randomization of the pens. corrected as suggested (L99-100)

L82-83: 5 replicates per treatment seems too low. What was the rationale for using this low replication? 

Based on a mixed feeding strategy, we selected 5 replications in 8 main groups with a balanced sex distribution for this study (mean of 7 females and 7 males per replication). This distribution was uniform within the henhouse. If we had increased the number of repetitions, we would have had to reduce the number of animals, which would have been problematic due to the possibility of animal deaths. In addition, there would be issues with the repetitions' uniform distribution. For this reason, we chose to work with five replicates, which were approved by our local animal welfare ethics committee. We believe that 5 repetitions per group is not less. The basis of this opinion is the studies carried out with this planning (given in the attached list). Moreover, there are studies with groups with less than 5 repetitions (given in the attached list)

5 replicates

  • Manyelo TG, Sebola NA, Mabelebele M. Effect of amaranth leaf meal on performance, meat, and bone characteristics of Ross 308 broiler chickens. PLoS One. 2022 Aug 9;17(8):e0271903. doi: 10.1371/journal.pone.0271903. PMID: 35944048; PMCID: PMC9362934.
  • Zarghi H,  Golian A,  Hassanabadi A, Khaligh F Effect of zinc and phytase supplementation on performance, immune response, digestibility and intestinal features in broilers fed a wheat-soybean meal diet, Italian Journal of Animal Science 2022, Vol. 21, No. 1, 430-444. https://doi.org/10.1080/1828051X.2022.2034061
  • Ahmadi-Sefat AA, Taherpour K, Ghasemi HA, Akbari Gharaei M, Shirzadi H, Rostami F. Effects of an emulsifier blend supplementation on growth performance, nutrient digestibility, intestinal morphology, and muscle fatty acid profile of broiler chickens fed with different levels of energy and protein. Poult Sci. 2022 Nov;101(11):102145. doi: 10.1016/j.psj.2022.102145. Epub 2022 Aug 28. PMID: 36155885; PMCID: PMC9519631.
  • Biesek J, Banaszak M, Wlaźlak S, Adamski M. The effect of partial replacement of milled finisher feed with wheat grains on the production efficiency and meat quality in broiler chickens. Poult Sci. 2022 May;101(5):101817. doi: 10.1016/j.psj.2022.101817. Epub 2022 Feb 25. PMID: 35339933; PMCID: PMC8960948.
  • Zhang S, Gong R, Zhao N, Zhang Y, Xing L, Liu X, Bao J, Li J. Effect of intermittent mild cold stimulation on intestinal immune function and the anti-stress ability of broilers. Poult Sci. 2023 Feb;102(2):102407. doi: 10.1016/j.psj.2022.102407. Epub 2022 Dec 9. PMID: 36571877; PMCID: PMC9803957.
  • Tavaniello S, Fatica A, Palazzo M, Zejnelhoxha S, Wu M, Marco L, Salimei E, Maiorano G. Carcass and Meat Quality Traits of Medium-Growing Broiler Chickens Fed Soybean or Pea Bean and Raised under Semi-Intensive Conditions. Animals (Basel). 2022 Oct 20;12(20):2849. doi: 10.3390/ani12202849. PMID: 36290235; PMCID: PMC9597835.
  • Lee WD, Kothari D, Moon SG, Kim J, Kim KI, Ga GW, Kim YG, Kim SK. Evaluation of Non-Fermented and Fermented Chinese Chive Juice as an Alternative to Antibiotic Growth Promoters of Broilers. Animals (Basel). 2022 Oct 12;12(20):2742. doi: 10.3390/ani12202742. PMID: 36290128; PMCID: PMC9597775.

4 replicate

  • Orczewska-Dudek S, Pietras M. The Effect of Dietary Camelina sativaOil or Cake in the Diets of Broiler Chickens on Growth Performance, Fatty Acid Profile, and Sensory Quality of Meat. Animals (Basel). 2019 Sep 27;9(10):734. doi: 10.3390/ani9100734. PMID: 31569656; PMCID: PMC6826988.
  • Kim HJ, Son J, Jeon JJ, Kim HS, Yun YS, Kang HK, Hong EC, Kim JH. Effects of Photoperiod on the Performance, Blood Profile, Welfare Parameters, and Carcass Characteristics in Broiler Chickens. Animals (Basel). 2022 Sep 3;12(17):2290. doi: 10.3390/ani12172290. PMID: 36078010; PMCID: PMC9454977.
  • Vasilopoulos S, Giannenas I, Savvidou S, Bonos E, I. Rumbos C, Papadopoulos E, Fortomari Ps, Christos G. Athanassiou, Growth performance, welfare traits and meat characteristics of broilers fed diets partly replaced with whole Tenebrio molitor larvae, Animal Nutrition, 2022, ISSN 2405-6545, https://doi.org/10.1016/j.aninu.2022.12.002.
  • Jerine A.J. van der Eijk, Jan van Harn, Henk Gunnink, Stephanie Melis, Johan W. van Riel, Ingrid C. de Jong, Fast- and slower-growing broilers respond similarly to a reduction in stocking density with regard to gait, hock burn, skin lesions, cleanliness, and performance, Poultry Science, Volume 102, Issue 5, 2023, 102603, ISSN 0032-5791, https://doi.org/10.1016/j.psj.2023.102603.

L96: remove "including" corrected as suggested

L116-119: It's not clear if 10 chickens were sacrificed for each day 21 and 28 42. corrected as suggested (L141-143)

L120-130: It's not clear if ad when you vaccinated them with IBD. Please specify corrected as suggested (L108-113)

Reviewer 4 Report

This study was conducted to determine the effects of dietary supplementation of sodium nitroprusside (SNP), a nitric oxide (NO) donor, and NG-nitro-L-arginine methyl ester (L-NAME), a NO synthase inhibitor, on growth performance, organ development, and immunity in broilers. The Introduction chapter provides an overview of the world's knowledge on this subject. The material used in the research is sufficiently numerous, and the research methods used are correct. The results are described correctly. The discussion is exhaustive. Summary of the results are correct. Some corrections are needed before publishing an article in Animals. The proposed changes are listed below.

L23 improved FCR? only 0-21 d

L86 or how many hours of light, what intensity (in lux), what color; what relative humidity during the rearing period

L152-154 Description inconsistent with data in Table 2

In Table 2: L-NAME25; L-NAME50; L-NAME100 instead of current form

L179 on day 21 not days 0-21

In Table 3 for Bursa of Fabricius, SNP 100 and 200, Superscript for 'b'

L409 Revue Med. Vet. 2010, 161, 409-417.

Author Response

We authors are grateful to the referee to improve our manuscript. The point-by-point comments have been explained as mentioned below.

This study was conducted to determine the effects of dietary supplementation of sodium nitroprusside (SNP), a nitric oxide (NO) donor, and NG-nitro-L-arginine methyl ester (L-NAME), a NO synthase inhibitor, on growth performance, organ development, and immunity in broilers. The Introduction chapter provides an overview of the world's knowledge on this subject. The material used in the research is sufficiently numerous, and the research methods used are correct. The results are described correctly. The discussion is exhaustive. Summary of the results are correct. Some corrections are needed before publishing an article in Animals. The proposed changes are listed below.

L23 improved FCR? only 0-21 d;  corrected as suggested (L23-24)

L86 or how many hours of light, what intensity (in lux), what color; what relative humidity during the rearing period. It has been added to the main text as follows. (L103-106)

During the experiment, the relative humidity was between 45% and 65%. For the first four days following placement, light (fluorescent, 30 lux) was provided for 23 hours, and it was gradually reduced (1 hour per day) to 20 hours on day 7 (fluorescent, 10 lux).

L152-154 Description inconsistent with data in Table 2;  corrected as suggested

In Table 2: L-NAME25; L-NAME50; L-NAME100 instead of current form

L179 on day 21 not days 0-21;  corrected as suggested. (L202)

In Table 3 for Bursa of Fabricius, SNP 100 and 200, Superscript for 'b': corrected as suggested. 

L409 Revue Med. Vet. 2010, 161, 409-417. corrected as suggested. (L443-444)
